# Reinforcement Learning Improves Traversal of Hierarchical Knowledge in LLMs

## Abstract

Reinforcement learning (RL) is often credited with improving language-model reasoning and generalization, possibly at the expense of memorized knowledge degradation. We observe, however, that on certain tasks designed to test pure knowledge recall; e.g., "Which disease corresponds to ICD-9 code 57.95?", RL-enhanced reasoning models (Deepseek-R1, QwQ, Magistral) still consistently surpass their non-reasoning counterparts (Deepseek-V3, Qwen-Instruct, Mistral-Small) by a large margin of 21 percentage points. Our analysis indicates that these gains stem not from acquisition of new knowledge during RL, but from improved access to knowledge already encoded during pretraining: RL appears to teach models to efficiently traverse hierarchical structure in the data to recall relevant information at inference time. To test this hypothesis we demonstrate that structured prompting designed to explicitly instruct for similar step-by-step hierarchy traversal recovers most of the RL gains, reducing the 21 pp gap to 6.1 pp on MedConceptsQA, without any RL training. Taken together, these results suggest that many benefits attributed to "reasoning training" may, in fact, arise from enhanced knowledge navigation rather than improved logical capability.

## 1 Introduction

Reinforcement Learning from Human Feedback (RLHF) and related techniques are widely understood to enhance Large Language Models' (LLMs) reasoning and generalization capabilities (OpenAI, 2024; Bai et al., 2022). This interpretation has motivated extensive research into RL-enhanced models like OpenAI's o1 (Jaech et al., 2024) and DeepSeek's R1 (Guo et al., 2025), which are marketed primarily for their superior performance on complex reasoning tasks. The conventional wisdom holds that these improvements come at a cost: reduced factual accuracy and memorization capabilities, often referred to as the "alignment tax" (Lin et al., 2024; Askell et al., 2021).

However, our empirical observations challenge this narrative. When evaluating reasoning-enhanced models (QwQ, R1, Magistral) against their base counterparts (Qwen-Instruct, Deepseek-V3, Mistral-Small) on MedConceptsQA—a benchmark consisting purely of medical code lookup tasks requiring no logical reasoning—we find that reasoning models consistently and substantially outperform base models. For instance, Deepseek-R1 achieves a 77.0% accuracy compared to Deepseek-V3's 56.0%, a striking 21 percentage point gap. This performance advantage is also observed with increasing task complexity: on hierarchical navigation tasks requiring 5 or more retrievals, reasoning model QwQ-32B achieve 67.1% accuracy compared to the 55.6% accuracy seen in its base counterpart. This pattern holds across model families, with reasoning variants consistently producing more accurate hierarchical paths especially as retrieval depth increases. This is surprising: if RL training primarily enhances reasoning at the expense of memorization, why would reasoning models excel at pure memorization tasks?[1]

Hierarchical structures are prevalent in biomedicine and genomics, where structured knowledge is essential for standardizing electronic health records, and descriptions of clinical diseases (Donnelly et al., 2006). In addition, hierarchical taxonomies observed in the biological processes, molecular functions and cellular components of gene products enable high-throughtput biomedical research (Consortium, 2019). Medical coding systems like ICD-9 and ICD-10 exemplify such hierarchical

---

[1]Throughout this work, we use "reasoning-enhanced" and "RL-enhanced" interchangeably, as the exact training methodology is not always transparent across models, but they share the characteristic of extended chain-of-thought reasoning capabilities.

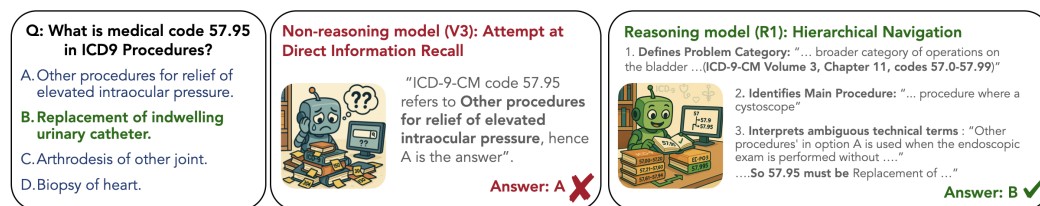

Figure 1: A sample response from a reasoning model (R1) and non-reasoning model (V3) on a memorization heavy look up task: Contrasting approaches to hierarchical knowledge retrieval on MedConceptsQA. Base models attempt direct recall and fail, while reasoning-enhanced models naturally reconstruct the knowledge hierarchy before systematic elimination, achieving 21pp higher accuracy.

structures, with parent codes representing general conditions and increasingly specific child codes detailing particular diagnoses. Successfully retrieving information in this case requires efficiently traversing these hierarchies. We therefore posit that the superior performance attained through RL on these tasks reveals a fundamental misunderstanding of how RL affects model capabilities; rather than creating new knowledge or improving logical deduction, RL training may primarily enhance models' ability to navigate and access their existing parametric knowledge—especially when that knowledge contains a hierarchical structure. The key insight comes from extensive empirical examinations of how these models approach the tasks differently. When given the medical code lookup task, reasoning-enhanced models first reconstruct the hierarchical structure of the knowledge domain (e.g., first stating "ICD-9 code 46.xx belongs to operations on the intestine", and then enumerating subcategories), then systematically eliminate incorrect options. In contrast, base and instruction-tuned models attempt to directly recall information.

If this hypothesis is correct, it has two important implications. First, it challenges the assumption that reasoning-enhanced models provide fundamentally new capabilities over their base counterparts; their superior performance may often stem from more effective knowledge recall rather than enhanced reasoning. Second, it suggests that structured prompting approaches - which explicitly instruct models to reconstruct hierarchies and perform stepwise elimination - could potentially achieve similar benefits without the computational expense of RL training.

We also examine distilled reasoning models (e.g., R1-Distill-Qwen-32B) to understand whether the navigation capability can be transferred through distillation. Our results reveal that distilled models achieve neither the performance of their reasoning counterparts nor that of well-prompted base models. For instance, R1-Distill-Qwen-32B scores 37.5% on MedConceptsQA (Template 1) compared to R1's 77.0%, but also falls short of both QwQ-32B (48.2%) and even Qwen2.5-32B with structured prompting (Template 3: 40.4%). This suggests that distillation not only fails to transfer the navigation capability effectively but may also degrade the model's access to its existing knowledge - a lose-lose scenario that questions the utility of reasoning distillation for knowledge-intensive tasks.

Our experiments reveal consistent patterns where reasoning models outperform on structured retrieval tasks, with carefully designed prompts recovering substantial portions of the performance gap. We additionally observe several interesting trends across model scales and specialized variants: math-specialized models (e.g., Qwen2.5-Math-7B: 20.9%) show degraded performance compared to their general counterparts (Qwen2.5-7B: 27.6%), suggesting that specialization may harm hierarchical knowledge navigation.

Finally, to test whether the necessary hierarchical knowledge exists in base models and only the navigation strategy differs, we craft structured prompts that guide models through the same systematic elimination and hierarchical traversal that reasoning models perform naturally. We find that structured prompting can close much of the performance gap, achieving up to 70% of the gains from RL training through prompting alone on MedConceptsQA and reducing a 21pp gap to 6.1pp (comparing Deepseek-V3 Template 3: 69.8% to R1 Template 1: 77.0%). Overall, we make the following contributions:

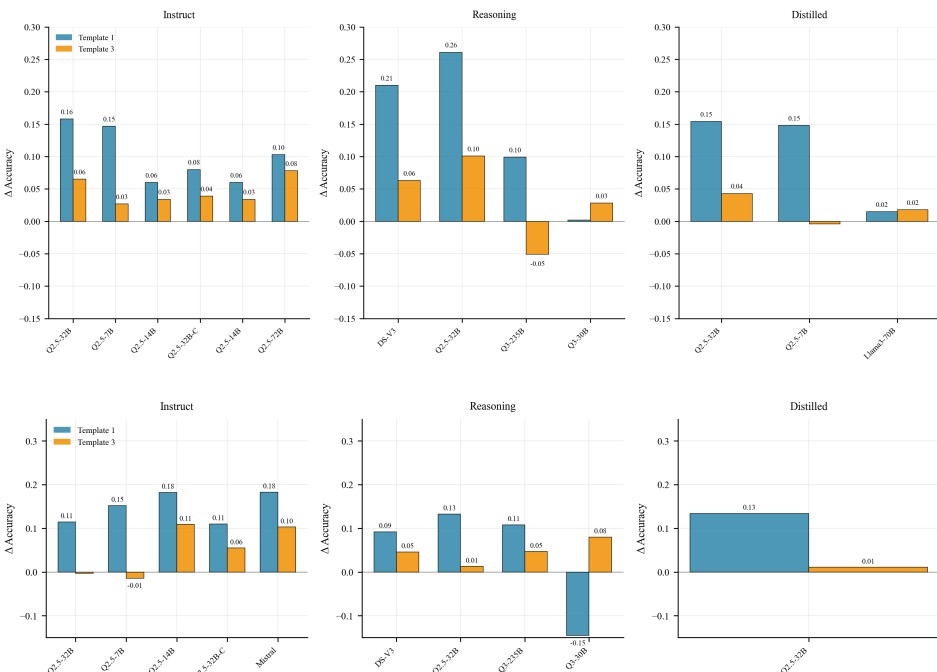

Figure 2: **Structured prompting reduces performance gaps across model enhancement techniques.** Performance differential ($\Delta = \text{Acc}_{\text{advanced}} - \text{Acc}_{\text{base}}$) between enhanced and base models on (a) MedConceptsQA and (b) IPC datasets. Three categories of model pairs are evaluated: *Instruction-tuned* (Qwen2.5 variants at 7B/14B/32B/72B scales, Qwen2.5-32B-Coder, Mistral-Small-Instruct), *Reasoning-enhanced* (Deepseek-V3→R1, Qwen2.5-32B→QwQ-32B, Qwen3-30B-A3B→Thinking, Qwen3-235B-A22B→Thinking), and *Distilled* (Qwen2.5-7/32B→R1-Distill-Qwen-7/32B, Llama3.3-70B-Instruct→R1-Distill-Llama-70B). Template 3 (orange) consistently reduces or reverses performance advantages compared to Template 1 (blue) across both datasets. Notable findings include: Deepseek-V3 gaining 15pp on IPC codes with Template 3, Qwen3-235B-Instruct outperforming its Thinking variant by 5pp, and Qwen2.5-7B surpassing its instruction-tuned counterpart on MedConceptsQA. These results demonstrate that prompt engineering can effectively substitute for specialized training across diverse reasoning tasks.

- We identify and document the surprising phenomenon of reasoning models outperforming base models on pure memorization tasks, challenging conventional assumptions about RL's effects

- We provide empirical evidence that RL's benefits in these contexts stem from enhanced knowledge navigation rather than knowledge creation or logical reasoning improvements, with distillation experiments confirming this mechanism

- We demonstrate that structured prompting can substitute for RL training on hierarchical retrieval tasks, with important implications for practical deployment.

- To effectively quantify performance with respect to hierarchical traversal, we introduce a new metric called the Hierarchical Alignment Score (HAS). When dividing tasks by retrieval complexity, we show that RL-led improvements in reasoning models fundamentally strengthen the model's ability to perform procedural knowledge traversal, as seen in the improved HAS score and accuracy.

## 2 TARGETED MEMORIZATION-HEAVY TASKS AND STRUCTURED PROMPTING

### 2.1 INFORMATION RECALL TASKS

To expand on our hypothesis that knowledge recall abilities differ between base and reasoning models, we consider tasks that can be classified as pure information recall (without added computation or

Table 1: Performance comparison across prompt templates on the MedConceptsQA dataset. Models are evaluated using three distinct prompt templates (T1-T3). Results show majority voting accuracy and mean accuracy ± standard deviation. Bold values indicate best performance within each model family.

| Model | Template 1 | | Template 2 | | Template 3 | |
|---|---|---|---|---|---|---|
| | Majority | Mean ± Std | Majority | Mean ± Std | Majority | Mean ± Std |
| *Deepseek Models* | | | | | | |
| Deepseek-V3 | 0.541 | 0.551 ± .014 | 0.632 | 0.636 ± .049 | 0.717 | 0.701 ± .026 |
| Deepseek-V3.1-Reasoning | **0.794** | 0.785 ± .009 | **0.805** | 0.772 ± .007 | 0.774 | 0.749 ± .005 |
| Deepseek-R1 | 0.778 | **0.83 ± .006** | 0.790 | **0.774 ± .013** | 0.792 | **0.775 ± .026** |
| *Qwen3 Family* | | | | | | |
| Qwen3-30B-A3B-Instruct-2507 | 0.439 | 0.425 ± .010 | 0.459 | 0.448 ± .006 | 0.474 | 0.466 ± .007 |
| Qwen3-30B-A3B-Thinking-2507 | 0.441 | 0.402 ± .010 | 0.518 | 0.498 ± .003 | 0.502 | 0.476 ± .007 |
| Qwen3-Coder-30B-A3B-Instruct | 0.448 | 0.408 ± .005 | 0.410 | 0.382 ± .012 | 0.476 | 0.463 ± .002 |
| Qwen3-Coder-480B-A35B-Instruct | 0.616 | 0.575 ± .005 | 0.567 | 0.562 ± .003 | **0.656** | **0.643 ± .004** |
| Qwen3-235B-A22B-Instruct-2507 | 0.542 | 0.503 ± .004 | 0.548 | 0.528 ± .005 | 0.631 | 0.589 ± .007 |
| Qwen3-235B-A22B-Thinking-2507 | **0.641** | **0.599 ± .003** | **0.656** | **0.617 ± .003** | 0.580 | 0.554 ± .008 |
| *Qwen2.5 7B Family* | | | | | | |
| Qwen2.5-7B | 0.148 | 0.159 ± .007 | 0.277 | 0.239 ± .036 | 0.286 | 0.270 ± .012 |
| Qwen2.5-7B-Instruct | 0.295 | 0.289 ± .006 | **0.329** | **0.316 ± .008** | **0.313** | **0.307 ± .015** |
| Qwen2.5-Coder-7B | 0.166 | 0.176 ± .007 | 0.259 | 0.245 ± .010 | 0.271 | 0.269 ± .006 |
| Qwen2.5-Math-7B | 0.209 | 0.215 ± .018 | 0.120 | 0.106 ± .026 | 0.265 | 0.258 ± .015 |
| Deepseek-R1-Distill-Qwen-7B | **0.296** | **0.292 ± .010** | 0.256 | 0.250 ± .017 | 0.282 | 0.289 ± .017 |
| *Qwen2.5 14B Family* | | | | | | |
| Qwen2.5-14B | 0.335 | 0.316 ± .015 | 0.332 | 0.293 ± .025 | 0.386 | 0.372 ± .007 |
| Qwen2.5-14B-Instruct | **0.395** | **0.385 ± .006** | **0.420** | **0.415 ± .007** | **0.420** | **0.409 ± .012** |
| Qwen2.5-Coder-14B | 0.314 | 0.301 ± .004 | 0.280 | 0.267 ± .004 | 0.309 | 0.288 ± .005 |
| Qwen2.5-Coder-14B-Instruct | 0.274 | 0.258 ± .006 | 0.265 | 0.248 ± .010 | 0.255 | 0.257 ± .014 |
| *Qwen2.5 32B Family* | | | | | | |
| Qwen2.5-32B | 0.221 | 0.219 ± .012 | 0.332 | 0.260 ± .071 | 0.404 | 0.372 ± .007 |
| Qwen2.5-32B-Instruct | 0.379 | 0.371 ± .012 | 0.475 | 0.449 ± .010 | 0.469 | 0.454 ± .007 |
| Qwen2.5-Coder-32B | 0.291 | 0.267 ± .012 | 0.309 | 0.264 ± .039 | 0.342 | 0.335 ± .007 |
| Qwen2.5-Coder-32B-Instruct | 0.371 | 0.345 ± .009 | 0.399 | 0.394 ± .003 | 0.381 | 0.371 ± .003 |
| Deepseek-R1-Distill-Qwen-32B | 0.375 | 0.351 ± .009 | 0.380 | 0.369 ± .005 | 0.447 | 0.420 ± .002 |
| QwQ-32B | **0.482** | **0.470 ± .012** | **0.513** | **0.487 ± .009** | **0.505** | **0.481 ± .005** |
| *Qwen2.5 72B Family* | | | | | | |
| Qwen2.5-72B | 0.443 | 0.389 ± .005 | 0.351 | 0.305 ± .028 | 0.468 | 0.418 ± .008 |
| Qwen2.5-72B-Instruct | **0.546** | **0.519 ± .007** | **0.520** | **0.512 ± .005** | **0.546** | **0.537 ± .008** |
| *Llama Models* | | | | | | |
| Llama3.3-70B-Instruct | 0.522 | **0.521 ± .004** | 0.525 | 0.525 ± .003 | 0.592 | 0.580 ± .005 |
| Deepseek-R1-Distill-Llama-70B | **0.537** | 0.495 ± .002 | **0.633** | **0.609 ± .011** | **0.610** | **0.596 ± .012** |
| *Distilled Models* | | | | | | |
| Deepseek-R1-Distill-Llama-8B | 0.362 | 0.339 ± .016 | 0.320 | 0.310 ± .010 | 0.364 | 0.343 ± .009 |
| *Mistral Family* | | | | | | |
| Mistral-Small-24B-Base-2501 | 0.227 | 0.233 ± .007 | 0.150 | 0.125 ± .043 | 0.327 | 0.296 ± .000 |
| Mistral-Small-3.1-24B-Base-2503 | **0.370** | **0.329 ± .013** | 0.267 | 0.214 ± .040 | 0.449 | 0.423 ± .014 |
| Magistral-Small-2507 | 0.300 | 0.287 ± .047 | **0.500** | **0.461 ± .018** | **0.482** | **0.463 ± .004** |

logical deduction). We select the following datasets, which contain Q&A that can be classified this way:

**MedConceptsQA:** A multiple-choice question answering dataset focused on biomedical and clinical concepts. The questions are designed to test factual recall of medical terminology, concept definitions, and their relationships, without requiring reasoning over patient cases or performing calculations.

**IPC:** The International Patent Classification dataset consists of queries mapped to patent classification codes. The task requires identifying the correct category for a given technical description, relying primarily on recall of standardized knowledge of patent domains rather than multi-step reasoning.

The questions are categorized by the depth of hierarchical traversal required, from simple lookups to deep-level ancestry and relationship detection. The difficulty level of each question type is defined by the number of hierarchical retrievals required:

- Memory-Light (ML) questions require 0-2 hierarchical retrievals and include basic structural understanding tasks such as Level Decoding, Parent Lookup, and Grandparent Lookup (for detail, refer to A.3). These questions test immediate recall of the patent classification structure and direct parent-child relationships.

- Memory-Moderate (MM) questions involve 3-4 hierarchical retrievals, including Great-grandparent Lookup tasks that require tracing multiple levels up the classification hierarchy while maintaining accurate recall of intermediate nodes.

- Memory-Heavy (MH) questions demand 5+ hierarchical retrievals and include complex relational tasks such as Cousin Relationship and Deepest Descent (see A.3).

In addition, the Common Ancestor task complexity varies based on the depth of codes involved, requiring models to trace multiple paths through the hierarchy and identify convergence points. For instance, finding the nearest common ancestor between two neighboring nodes requires light memory recall, whereas distant classification codes may require traversing 5-6 levels for each code path.

To investigate how reasoning versus base models' hierarchical knowledge capabilities change as retrieval depth increases, we further stratify the Common Ancestor task into these complexity levels to compare performance difference.

**MMLU:** The Massive Multitask Language Understanding benchmark covers 57 academic and professional subjects (e.g., history, law, medicine, physics). Each question is multiple-choice and tests factual knowledge across diverse domains. We restrict evaluation to the *Prehistory* subject area. This subset consists of multiple-choice questions assessing factual knowledge of early human history, archaeology, and ancient civilizations. Since the questions primarily test recall of historical facts, they align well with our definition of information recall.

## 2.2 Hierarchical Alignment Score (HAS)

To further evaluate the quality of hierarchical path predictions in IPC codes, we devise a novel metric, the Hierarchical Alignment Score (HAS), through two key components:

**Alignment F1-Score** ($AF_1$)**:** Conceptually, this measures the balance between precision and recall for hierarchical ancestor identification. It is mathematically defined as:

$$AF_1 = \frac{2 \times \text{AP} \times \text{AR}}{\text{AP} + \text{AR}} \tag{1}$$

where AP (Alignment Precision) and AR (Alignment Recall) follow standard precision and recall definitions applied to hierarchical ancestor sets (Buckland & Gey, 1994).

**Common Subsequence Score** ($CSS$)**:** This evaluates how well the sequential hierarchical structure is preserved, using the Longest Common Subsequence (Paterson & Dančík, 1994) as:

$$CSS = \frac{\text{Length of the Longest Common Subsequence}}{\text{Total number of ancestors in ground truth path}} \tag{2}$$

The final HAS score combines these components through a harmonic mean:

$$\text{HAS} = \frac{2 \times AF_1 \times \text{CSS}}{AF_1 + \text{CSS}} \tag{3}$$

This evaluates a model's ability to maintain structural coherence while navigating the patent classification hierarchy.

## 2.3 Prompting approaches

We evaluate model performance on MedConceptsQA and IPC using three different prompt templates. Template 1 (Baseline Answer Only) requires a direct, single-letter answer without any explanations. Template 3 (Structured Hierarchical Navigation) tests hypothesis by instructing the model to perform a structural breakdown of the main concept in the query, followed by stepwise elimination of incorrect

options. We also include Template 2 (Standard Chain-of-Thought), which asks for an answer with an explanation, and a fully open-ended variant for diagnostic purposes (see Appendix A.1 for detail). Our analysis in Figures 2 and 3 focuses primarily on the comparison between Template 1 and Template 3, as Template 2 offers only marginal gains over the original prompt.

We evaluated each model on three prompt templates over three independent runs, using a sampling temperature of 0.8 and a top-p of 0.7. Performance is reported as both the mean accuracy (± standard deviation) and the majority-voted accuracy across the runs.

## 3 EXPERIMENTAL RESULTS

We make the following observations:

**Observation 1: Specialized training such as RL reinforces strategic reasoning over explicit knowledge recall.** When evaluated on open-ended queries where multiple-choice options from MedConceptsQA are withheld, both base (e.g., Deepseek-V3) and their reasoning counterparts (e.g., R1) exhibit a lack of direct knowledge recall. [2] This failure occurs even for questions that the reasoning model correctly answers when presented with a set of options (see Appendix A.2). This finding suggests that RL training reinforces procedural reasoning strategies, such as hierarchical recall and systematic elimination, which enable a model to identify the correct answer from a provided set rather than retrieving it from memorized knowledge.

**Observation 2: By adopting strategies such as hierarchical navigation and stepwise elimination into prompts, the accuracy gap between base models and their instruction-tuned, or RL-tuned variants narrow.** On MedConceptsQA, the 23.7 pp accuracy difference between Deepseek-R1 (77.8%) and its base model V3 (54.1%) under a simple prompt (Template 1) shrinks to just 7.5pp gap when the base model is guided by a structured prompt (Template 3) (see Table 1 for detail). On the IPC codes, a structured prompt makes Qwen2.5-32B's accuracy increase from 64.4% to 77.7%, surpassing its instruction-tuned model (77.4%). From Figures 2 and 3, we can see sometimes the accuracy of base models can surpass that of more advanced variants when switching to Template 3. For instance, on MedConceptsQA, the base Qwen3-235B-A22B-Instruct achieves 63.1% accuracy with Template 3, outperforming its reasoning-enhanced counterpart, Qwen3-235B-A22B-Thinking, which scores 58.0% on the same prompt.

**Observation 3: Distilling knowledge from larger teacher models into smaller ones does not make the student models possess hierarchical reasoning capabilties comparable to those of teacher models.** On MedConceptsQA, the student model Deepseek-R1-distilled-Qwen32B achieves a majority-voting accuracy (T1: 37.5%), significantly lower than its teacher model Deepseek-R1 (T1: 77.8%). Even though the distilled model gives better performance than its base model Qwen2.5-32B, it is outperformed by the reasoning model of the same size, QwQ-32B, by a large margin (48.2%). Similar behaviour is also seen on IPC codes, where the distilled model (77.8%) again underperforms its teacher (92.3%). This suggests that direct distillation is a less effective method to promote strategic reasoning.

**Observation 4: Reasoning models excel on tasks requiring deeper hierarchical traversal, although performance on simpler tasks can be more nuanced.** As shown in Table 3, while the Qwen2.5-32B base model and its reasoning counterpart QwQ-32B have similar performance on simpler tasks (e.g., both achieving 33.7% accuracy on Memory-Moderate tasks), a significant gap emerges with increasing complexity. For Memory-Heavy tasks, the QwQ-32B model achieves a higher accuracy of 67.1% compared to the base model's 55.6%. Crucially, the HAS score for the QwQ model is also higher on Memory-Heavy tasks (0.430 vs. 0.405), indicating a better quality hierarchical path.

This trend is consistent with the Deepseek family. While Deepseek-V3 achieves higher accuracy on Memory-Medium tasks (45.0% vs. 32.5%), both it and Deepseek-R1 reach an equal 67.7% on Memory-Heavy tasks. Additionally, R1 maintains a significantly higher HAS score (0.597 vs. 0.503), indicating a superior quality of hierarchical path traversal. This pattern is even more evident in the Mistral family, where the reasoning counterpart Magistral has the highest accuracy and HAS score

---

[2]While this is often achieved via Reinforcement Learning (RL), the exact training methodologies are frequently not disclosed. Our observations thus apply to the outcome of this specialized training rather than a specific, confirmed technique.

Table 2: Performance comparison across prompt templates on the IPC codes. Models are evaluated using three distinct prompt templates (T1-T3). Results show majority voting accuracy and mean accuracy ± standard deviation. Bold values indicate best performance within each model family.

| Model | Template 1 | | Template 2 | | Template 3 | |
|---|---|---|---|---|---|---|
| | Majority | Mean ± Std | Majority | Mean ± Std | Majority | Mean ± Std |
| *Deepseek Models* | | | | | | |
| Deepseek-V3 | 0.831 | 0.846 ± .000 | **0.923** | **0.882 ± .007** | 0.877 | 0.872 ± .007 |
| Deepseek-R1 | **0.923** | **0.913 ± .019** | 0.892 | 0.867 ± .026 | **0.923** | **0.903 ± .007** |
| *Qwen3 Family* | | | | | | |
| Qwen3-30B-A3B-Instruct-2507 | 0.682 | 0.682 ± .015 | 0.708 | 0.708 ± .022 | 0.749 | 0.749 ± .007 |
| Qwen3-30B-A3B-Thinking-2507 | 0.537 | 0.451 ± .019 | 0.572 | 0.467 ± .032 | 0.829 | 0.621 ± .036 |
| Qwen3-Coder-30B-A3B-Instruct | 0.682 | 0.682 ± .007 | 0.646 | 0.646 ± .013 | 0.697 | 0.697 ± .007 |
| Qwen3-235b-a22b-Instruct-2507 | 0.800 | 0.800 ± .013 | 0.846 | 0.846 ± .013 | 0.846 | 0.846 ± .013 |
| Qwen3-235b-a22b-Thinking-2507 | **0.908** | **0.846 ± .013** | **0.877** | **0.836 ± .026** | **0.893** | **0.851 ± .015** |
| *Qwen2.5 7B Family* | | | | | | |
| Qwen2.5-7B | 0.463 | 0.349 ± .040 | 0.436 | 0.364 ± .038 | **0.588** | **0.585 ± .038** |
| Qwen2.5-7B-Math | 0.386 | 0.303 ± .007 | 0.425 | 0.364 ± .038 | 0.411 | 0.354 ± .044 |
| Qwen2.5-7B-Instruct | **0.615** | **0.615 ± .025** | **0.554** | **0.554 ± .013** | 0.574 | 0.574 ± .015 |
| Qwen2.5-7B-Coder | 0.506 | 0.426 ± .036 | 0.520 | 0.467 ± .040 | 0.582 | 0.544 ± .032 |
| *Qwen2.5 14B Family* | | | | | | |
| Qwen2.5-14B | 0.526 | 0.421 ± .038 | 0.608 | 0.492 ± .033 | 0.609 | 0.600 ± .013 |
| Qwen2.5-14B-Instruct | **0.708** | **0.708 ± .025** | **0.691** | **0.687 ± .029** | **0.718** | **0.718 ± .007** |
| *Qwen2.5 Family (32B-72B)* | | | | | | |
| Qwen2.5-32B | 0.644 | 0.482 ± .059 | 0.641 | 0.503 ± .038 | 0.777 | 0.769 ± .013 |
| Qwen2.5-32B-Instruct | **0.759** | **0.759 ± .007** | 0.754 | 0.754 ± .000 | 0.774 | 0.774 ± .007 |
| Qwen2.5-32B-Coder | 0.627 | 0.595 ± .019 | 0.670 | 0.615 ± .055 | 0.740 | 0.687 ± .019 |
| Qwen2.5-32B-Coder-Instruct | 0.737 | 0.733 ± .015 | 0.718 | 0.718 ± .026 | **0.795** | **0.795 ± .007** |
| Qwen2.5-72B-Instruct | **0.759** | **0.759 ± .007** | **0.763** | **0.759 ± .019** | **0.795** | **0.795 ± .019** |
| Qwen2.5-72B-Math | 0.542 | 0.133 ± .019 | 0.542 | 0.164 ± .007 | 0.644 | 0.574 ± .015 |
| *Reasoning-Enhanced Models* | | | | | | |
| R1-Distill-Qwen-32B | **0.778** | **0.754 ± .038** | 0.730 | 0.667 ± .019 | 0.788 | 0.780 ± .019 |
| QwQ-32B | 0.777 | 0.713 ± .015 | **0.875** | **0.754 ± .070** | **0.790** | **0.769 ± .033** |
| *Mistral Family* | | | | | | |
| Mistral-Small-3.1-Base | 0.515 | 0.349 ± .007 | 0.654 | 0.436 ± .048 | 0.677 | 0.354 ± .033 |
| Mistral-Small-3.1-Instruct | 0.698 | 0.687 ± .026 | 0.679 | 0.662 ± .033 | 0.780 | 0.600 ± .033 |
| Magistral-Small-2507 | **0.801** | **0.744 ± .044** | **0.782** | **0.718 ± .007** | **0.801** | **0.744 ± .044** |

almost across all three retrieval levels. These together suggests that while base models can sometimes succeed, reasoning models consistently produce higher-quality, more accurate hierarchical paths, especially as the task complexity increases.

Our work demonstrates that reasoning models exhibit procedural reasoning strategies, such as hierarchical navigation and systematic elimination, rather than instilling explicit factual knowledge, as models that succeed on multiple-choice questions often fail the same queries in an open-ended format. Consequently, these accuracy gains can be replicated in base models through structured prompting that explicitly instructs these same strategies, narrowing the gap. However, this skill does not transfer via distillation, as student models perform significantly worse than their teachers, suggesting direct RL-tuning is a more effective enhancement method. The true benefit of RL-based enhancements becomes apparent when tasks are divided by retrieval complexity. As task complexity increases, reasoning models don't just achieve better accuracy, but fundamentally improve their procedural knowledge traversal, as shown by the Hierarchical Alignment Score (HAS).

## 4 RELATED WORK

### 4.1 THE ALIGNMENT TAX AND FACTUAL DEGRADATION

The trade-off between alignment and factual accuracy has been extensively explored. Lin et al. (2024) introduced the concept of the "alignment tax," demonstrating systematic performance degradation

Table 3: Model Performance on Common Ancestor Questions by Complexity

| 2*Model | ML (Memory-Light) | | MM (Memory-Medium) | | MH (Memory-Heavy) | |
|---|---|---|---|---|---|---|
| | Acc. (%) | HAS | Acc. (%) | HAS | Acc. (%) | HAS |
| *Deepseek Family* | | | | | | |
| Deepseek-V3 | 37.9 | 0.627 | **45.0** | 0.589 | **67.7** | 0.503 |
| Deepseek-R1 | **44.8** | **0.681** | 32.5 | **0.620** | **67.7** | **0.597** |
| *Qwen2.5-32B Family* | | | | | | |
| Qwen2.5-32B | **43.1** | **0.527** | **33.7** | 0.468 | 55.6 | 0.405 |
| Qwen2.5-32B-Instruct | 34.9 | 0.458 | 32.5 | **0.491** | 64.7 | **0.451** |
| QwQ-32B | 40.3 | 0.457 | **33.7** | 0.426 | **67.1** | 0.430 |
| *Mistral Family* | | | | | | |
| Mistral-Small-3.1-24B-Base-2503 | 28.6 | 0.349 | 36.2 | 0.382 | 27.9 | 0.303 |
| Mistral-Small-3.1-24B-Instruct-2503 | 37.4 | 0.610 | 37.3 | **0.548** | 64.0 | 0.522 |
| Magistral-Small-2507 | **66.7** | **0.723** | **50.0** | 0.536 | **87.5** | **0.569** |

on factual benchmarks as RLHF reward increases. Their analysis across six factual datasets (ARC-Easy, ARC-Challenge, RACE, PIQA, SQuAD, DROP) showed consistent deterioration with stronger alignment. OpenAI (2023) corroborated these findings, reporting that "RLHF does not improve exam performance (without active effort, it actually degrades it)" and noting reduced calibration after alignment.

Gudibande et al. (2023) provide mechanistic insights, showing that instruction tuning primarily teaches style rather than new knowledge, with models learning to respond confidently even when lacking relevant information. Recent work by Allen-Zhu & Li (2024) on the "Physics of Language Models" series demonstrates that base models often contain more accessible factual knowledge before alignment modifies retrieval patterns. Kassner et al. (2025) explicitly show that RLHF can "reverse memorization from supervised fine-tuning," supporting our observation that base models maintain superior raw factual recall.

## 4.2 REASONING ENHANCEMENT THROUGH RL

The dominant narrative positions RL as a reasoning amplification technique. Lightman et al. (2023) demonstrate how process supervision during RL training improves mathematical reasoning, while Wang et al. (2024c) show similar gains through step-wise reward models. Havrilla et al. (2024) formalize how RL teaches models to "think before they speak," developing internal reasoning chains.

However, recent work hints at a more nuanced picture. Zelikman et al. (2024) introduce Quiet-STaR, showing that training models to generate rationales improves downstream reasoning by teaching systematic exploration of solution spaces—essentially navigation skills. Shinn et al. (2023) demonstrate that reinforcement learning primarily helps models learn from feedback to refine their search through problem spaces, rather than acquiring new problem-solving rules. These findings align with our hypothesis that RL enhances navigation of existing knowledge structures.

## 4.3 HIERARCHICAL REASONING AND STRUCTURED NAVIGATION

Recent advances in hierarchical reasoning provide theoretical support for our knowledge navigation hypothesis. The Hierarchical Reasoning Model (HRM) (Ji et al., 2024) achieves near-perfect accuracy on complex tasks through interdependent modules enabling "hierarchical convergence"—precisely the type of structured traversal we require for tasks like medical code lookup. Wang et al. (2024b) show that hierarchical reinforcement learning on template sequences outperforms traditional chain-of-thought, suggesting RL teaches systematic navigation rather than new facts. In the medical domain, Choi et al. (2024) demonstrate that reasoning-augmented LLMs consistently outperform base models on ICD code classification, particularly at higher hierarchy levels. Zhang et al. (2024) report that Med-R1 achieves 29.94% improvement over base models through "structured medical reasoning," though they attribute this to reasoning rather than navigation enhancement.

## 4.4 PROMPTING AS AN ALTERNATIVE TO RL

The possibility of achieving RL-like benefits through prompting has gained increasing attention. Xu et al. (2024) demonstrate that Genetic-Evolution Prompt Alignment (GEPA) can outperform Group Relative Policy Optimization by up to 20% while using 35× fewer computational resources.

They argue that "the interpretable nature of language provides a richer learning medium than sparse scalar rewards." Wei et al. (2022) show that chain-of-thought prompting can match fine-tuned performance on reasoning tasks, while Zhou et al. (2023) demonstrate that optimized prompts can exceed supervised fine-tuning. Particularly relevant is work by Xu et al. (2023) showing that careful prompt engineering can close performance gaps between base and instruction-tuned models on structured tasks. The "Invisible Leash" phenomenon (Chen et al., 2024) reveals that much of RLHF's apparent benefit comes from teaching models to follow implicit formatting patterns—effects reproducible through prompting.

## 4.5 KNOWLEDGE STORAGE VERSUS KNOWLEDGE ACCESS

The distinction between knowledge acquisition and knowledge retrieval is crucial to our thesis. Wang et al. (2024a) demonstrate that fine-tuning rarely injects genuinely new facts, requiring extensive repetition for simple factual updates. Ovadia et al. (2024) show that models fine-tuned on new knowledge often "hallucinate" by incorrectly combining existing knowledge rather than storing new information. Dziri et al. (2024) provide key insights with their finding that models rely on "procedural knowledge extracted from documents involving similar reasoning processes" rather than memorizing new facts. This aligns with our hypothesis that RL enhances navigation strategies rather than expanding knowledge. Berglund et al. (2024) further support this through their "Reversal Curse" findings—models trained on "A is B" cannot infer "B is A," suggesting that training affects access patterns rather than creating bidirectional knowledge representations.

## 4.6 MEDICAL KNOWLEDGE SYSTEMS AND HIERARCHICAL LOOKUP

Medical coding systems provide an ideal testbed for distinguishing knowledge storage from navigation. The International Classification of Diseases (ICD) employs strict hierarchical structures where successful code identification requires systematic traversal (World Health Organization, 2019). Kraljevic et al. (2021) show that models struggle with medical codes not due to lack of exposure but difficulty navigating complex taxonomies. MedConceptsQA (MedConceptsQA Team, 2024) encompasses 800K+ medical concepts across ICD-9, ICD-10, CPT, and ATC classifications. Initial benchmarks showed most clinical LLMs performing near random chance despite extensive medical training, while general-purpose models with better structural reasoning capabilities significantly outperformed them.

This pattern—domain-specialized models losing to general models—suggests that the challenge lies in information navigation rather than possession. Our work synthesizes these disparate findings to argue that reasoning-enhanced models' superiority on memorization tasks stems from improved access to pre-existing knowledge rather than enhanced reasoning or expanded memory.

## 4.7 RETRIEVAL COMPLEXITY IN KNOWLEDGE-INTENSIVE TASKS

Recent work has begun to to examine the relationship between retrieval complexity and model performance in knowledge-intensive tasks. Gabburo et al. (2024) show that retrieval complexity extend beyond simple multi-hop reasoning—including temporal (15%), comparative (10%), and aggregate (16%) questions—suggesting that different types of knowledge organization require distinct retrieval strategies. Min et al. (2023) demonstrate that in long-form generation, factual accuracy in biographies drops as entity rarity increases, suggesting that retrieval difficulty directly impacts knowledge accessibility.

## 5 CONCLUSION

We make a counterintuitive observation that RL-enhanced models exhibit improve performance on tasks which require recalling memorized information, and posit that the improvements stem from the ability to better traverse hierarchical structures in the underlying data. This re-framing has significant implications for both our understanding of RL's effects, and can inform practical decisions about when RL training is necessary and when structured prompting suffices.

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

## A  TECHNICAL APPENDICES AND SUPPLEMENTARY MATERIAL

### A.1  ZERO-SHOT PROMPT TEMPLATES

We present four prompt templates used in MedConceptsQA and IPC, which are designed to elicit specific responses from language models. These templates request:

- Direct answers, both with and without explanations.

- Structural recall of codes and a stepwise elimination of incorrect options.

- An open-ended reasoning process that repeats the recall and elimination tasks but without providing any options.

---

**Prompt Template 1: MCQ with Final Answer Only**

Answer only A,B,C,D according to the answer to this multiple choice question.
*[... Insert Question Text Here ...]*
**Answer (only the letter of your choice (A, B, C, or D)):**

---

**Prompt Template 2: MCQ with Explanation**

You are a medical research assistant. Read the following multiple-choice question carefully. Your task is to:
1. Answer each question with one of A/B/C/D, which corresponds to the four options.
2. For my convenience, please give me a list of ANSWERs for the given instances in the format 'Answer: ...', with additional explanation for each answer in the format 'Explanation: ...'.
Respond in the following format:
Answer: <A/B/C/D>
Explanation: <your explanation here>

*[... Insert Question Text Here ...]*

**Answer:**
**Explanation:**

---

**Prompt Template 3: MCQ with Stepwise Reasoning**

You are a medical classification expert. For each option, first **recall the general category and structure breakdown of the medical code**, then explain **why it might be wrong**. Finally pick the correct one.

------

*[... Insert Question Text Here ...]*
**Steps to follow:**
1. Recall the general category and structural break down of the code.
2. Evaluate each option (A–D) briefly.
3. Choose the best option and justify.
**Answer format:**
Step 1: . . .
Step 2A: . . .
Step 2B: . . .
Step 2C: . . .
Step 2D: . . .
Final Answer: [A/B/C/D] because . . .

------

## Prompt Template 4: Open-Ended Question (No Options)

You are a medical classification expert. For each question, first **recall the general category and structure breakdown of the medical code**, then **state the final answer**.

*[... Insert Main Question Extracted from Text Here ...]*

**Steps:**
 1. Recall the general category and structural break down of the code.
 2. State your only final answer.

**Answer format:**
```
Step 1: ...
Step 2 Final Answer:
```

A.2 EXAMPLES ON MODEL RESPONSES UNDER VARIOUS PROMPTS

## Example: Medical Code Analysis (Question)

**Question:** What is the description of the medical code P02CX03 in ATC?
 A. moxidectin
 B. pyrvinium
 C. ANTINEMATODAL AGENTS
 D. bephenium

**Ground Truth Answer:** A
**Vocab:** ATC
**Level:** hard

## V3 Model Response Under Template 4 (Open-Ended, Incorrect)

STEP 1: CODE BREAKDOWN
The ATC code P02CX03 is part of the P02 category (anthelmintics), specifically under P02C (other anthelmintics). The "X" indicates a miscellaneous subgroup, and "03" is the specific substance.

STEP 2 FINAL ANSWER
tribendimidine

**Final Answer:** tribendimidine
**Correct:** False

## R1 Model Response Under Template 4 (Open-Ended, Incorrect),

STEP 1: CODE BREAKDOWN
The ATC code P02CX03 breaks down as follows:
 • **P**: Antiparasitic products, insecticides, and repellents
 • **P02**: Anthelmintics (drugs for parasitic worms)
 • **P02C**: Antitrematodal agents (targeting fluke infections)
 • **P02CX**: Other antitrematodal agents
 • **03**: Specific substance identifier for artemether.

STEP 2 FINAL ANSWER
artemether

**Final Consolidated Answer:** artemether
**Correct:** False

## V3 Model Response Under Template 3 (Incorrect)

STEP 1: RECALL THE GENERAL CATEGORY AND STRUCTURAL BREAKDOWN OF THE CODE

- **ATC Code Structure**:
  - **P**: Antiparasitic products, insecticides, and repellents (anatomical main group).
  - **P02**: Anthelmintics (therapeutic subgroup).
  - **P02C**: Antinematodal agents (pharmacological subgroup).
  - **P02CX**: Other antinematodal agents (chemical subgroup).
  - **P02CX03**: Pyrvinium (specific substance).

Thus, **P02CX03** is a code for **pyrvinium**, an antinematodal agent under "other antinematodal agents."

---

STEP 2: EVALUATE EACH OPTION

**A. moxidectin**
*Why it's wrong:* moxidectin is an antiparasitic drug, but its ATC code is **P02CX06**, not P02CX03.

**B. pyrvinium**
*Why it's correct:* matches the exact ATC code **P02CX03** for pyrvinium.

**C. ANTINEMATODAL AGENTS**
*Why it's wrong:* while **P02C** is the category for antinematodal agents, **P02CX03** is a specific drug (pyrvinium), not the entire group.

**D. bephenium**
*Why it's wrong:* bephenium is an antinematodal agent, but its ATC code is **P02CX01**, not P02CX03.

---

FINAL ANSWER: [B] BECAUSE P02CX03 CORRESPONDS SPECIFICALLY TO PYRVINIUM IN THE ATC CLASSIFICATION.

**Final Answer:** B
**Correct:** False

**R1 Model Response Under Template 3 (Correct)**

STEP 1: CODE BREAKDOWN

The ATC code **P02CX03** breaks down as follows:
- **P**: Antiparasitic products
- **P02**: Anthelmintics
- **P02C**: Antinematodal agents (3rd level)
- **P02CX**: Other antinematodal agents (4th level)
- **P02CX03**: Specific substance (5th level)

---

STEP 2: OPTION EVALUATION

**A. moxidectin**
*Evaluation:* Plausible as it is an anthelmintic, but its classification under P02CX03 must be confirmed.

**B. pyrvinium**
*Evaluation:* Is assigned to **P02CX01**, making this incorrect for CX03.

**C. ANTINEMATODAL AGENTS**
*Evaluation:* Refers to the 3rd level (**P02C**), not the 5th-level substance code.

**D. bephenium**
*Evaluation:* Is classified under **P02CA01** (tetrahydropyrimidine derivatives), not P02CX.

---

FINAL ANSWER: [A] BECAUSE P02CX03 CORRESPONDS TO MOXIDECTIN IN THE ATC CLASSIFICATION, WHILE OTHER OPTIONS ARE EITHER INCORRECT CODES (B, D) OR A CATEGORY-LEVEL DESCRIPTOR (C).

**Final Answer:** A
**Correct:** True

## A.3 IPC CODES EXAMPLES GROUPED BY DIFFERENT MEMORY RECALL LEVELS

### A.3.1 MEMORY-LIGHT (0-2 HIERARCHICAL RECALLS)

**Level Decoding:** This tests the ability to decode a hierarchical code (in this case, F02B 19/00) by identifying its constituent parts, such as Section, Class, Subclass, and Group.

**Memory-Light Questions - Level Decoding**

**Question:** In F02B 19/00, identify the Section, Class, Subclass, and Group.
**Options:**
    A) Section=F, Class=02, Subclass=B, Group=19/00
    B) Section=F, Class=2, Subclass=B, Group=19
    C) Section=B, Class=19, Subclass=F, Group=02/00
    D) Section=F, Class=02B, Subclass=19, Group=00
**Answer:** A
**Description:** Engines with precombustion chambers

**Parent Lookup:** This task requires the model to identify the parent of a given patent code. It is a Memory-Light task as it involves one memory recall to find the direct ancestor.

## Memory-Light Questions - Parent Lookup

**Question:** The immediate parent of F02B 1/04 is:
**Options:**
- A) F02B 1/00
- B) F02B 1/02
- C) F02B 1/06
- D) F02B

**Answer:** A
**Description:** Engines characterised by fuel-air mixture compression

**Grandparent Lookup:** This task requires the model to identify the second-level ancestor of a given patent code. It is a Memory-Light task as it involves a short, direct traversal up the hierarchy to find a specific ancestor.

## Memory-Light Questions - Grandparent Lookup

**Question:** Second-level ancestor of D01G 15/68 is:
**Options:**
- A) D01G 15/64
- B) D01G 15/46
- C) D01G 15/12
- D) D01G 15/00

**Answer:** B
**Reasoning:** D01G 15/68 → D01G 15/64 → D01G 15/46

**Sibling Discrimination:** This is a Memory-Light task that requires the model to identify a code that shares the same main group but has a different subgroup. It tests the model's ability to recognize and compare codes at a shallow hierarchical level.

## Memory-Light Questions - Sibling Discrimination

**Question:** Which is a sibling (same main group, different subgroup) of F02B 53/12?
**Options:**
- A) F02B 55/12
- B) F02B 53/00
- C) F02B 53/10
- D) F03B 53/06

**Answer:** C
**Description:** Ignition for rotary-piston engines

### A.3.2 MEMORY-MODERATE (3-4 HIERARCHICAL RECALLS)

**Great-grandparent Lookup:** This is a more challenging task that requires tracing a code's lineage back three levels to find the correct ancestor. Classified as a Memory-Moderate task, it tests the model's ability to handle slightly longer and more complex hierarchical paths.

## Memory-Moderate Questions - Great-grandparent Lookup

**Question:** Third-level ancestor of C01B 32/194 is:
**Options:**
- A) C01B 32/00
- B) C01B
- C) C01B 32/18
- D) C01B 32/19

**Answer:** A
**Reasoning:** C01B 32/194 → C01B 32/19 → C01B 32/18 → C01B 32/00

**Path Reconstruction:**  This Memory-Moderate task challenges the model to reconstruct the full descriptive name chain for a given patent code. It tests the model's ability to accurately recall and order the hierarchical labels (Section, Class, Subclass, etc.) that lead to a specific code.

---

**Memory-Moderate Questions - Path Reconstruction**

**Question:** Give the name chain for F02B 19/00 from Section → Class → Subclass → Main group

**Options:**
- A) Mechanical Engineering → Pumps → Piston Engines → Precombustion Chambers
- B) Mechanical Engineering → Combustion Engines → Piston Engines → Engines with precombustion chambers
- C) Lighting → Engines → Combustion Engines → Precombustion Chambers
- D) Mechanical Engineering → Combustion Engines → Gas Turbines → Precombustion Chambers

**Answer:** B

**Description:** Engines with precombustion chambers

---

A.3.3    MEMORY-HEAVY (5 OR MORE HIERARCHICAL RECALLS)

**Cousin Relationship:**  This is a Memory-Heavy task that tests the model's ability to understand lateral relationships within the hierarchy. It requires traversing up to a common grandparent and then back down to identify a "first cousin" that shares the same ancestor.

---

**Memory-Heavy Questions - Cousin Relationship**

**Question:** First cousin of H04N 9/806 (same grandparent level) is:

**Options:**
- A) H04N 9/808
- B) H04N 9/815
- C) H04N 9/82
- D) H04N 9/804

**Answer:** B

**Reasoning Paths:**
- H04N 9/806 → H04N 9/804 (parent) → H04N 9/80 (grandparent) → H04N 9/808 (uncle/aunt)
- H04N 9/806 → H04N 9/804 (parent)→ H04N 9/80 (grandparent)→ H04N 9/81 (uncle/aunt) → H04N 9/815 (cousin)
- H04N 9/806 → H04N 9/804 (parent) → H04N 9/80 (grandparent) → H04N 9/82 (uncle/aunt)
- H04N 9/806 → H04N 9/804 (parent)→ H04N 9/80 (grandparent) → H04N 9/804 (uncle/aunt)

---

**Deepest Descent:**    This Memory-Heavy question asks the model to identify the most specific descendant of a given patent code from a list of options. It tests the model's ability to perform deep, multi-step traversal down the hierarchy to determine which option has the longest, most specific path.

### Memory-Heavy Questions - Deepest Descent

**Question:** Most specific descendant of A01B 3/00 among these options:
**Options:**
    A) A01B 3/04
    B) A01B 3/426
    C) A01B 3/08
    D) A01B 3/26
**Answer:** B
**Reasoning Paths:**
- A01B 3/00 → A01B 3/04
- A01B 3/00 → A01B 3/36 → A01B 3/40 → A01B 3/42 → A01B 3/426
- A01B 3/00 → A01B 3/04 → A01B 3/06 → A01B 3/08
- A01B 3/00 → A01B 3/24 → A01B 3/26

**Orphan Detection:** As a Memory-Heavy task, this question asks the model to identify which of the given code pairs does not represent a valid parent-child relationship. This tests the model's deep knowledge of the hierarchical structure and its ability to spot inconsistencies.

### Memory-Heavy Questions - Orphan Detection

**Question:** Which does not represent a valid parent-child relationship?
**Options:**
    A) D01F 6/26 → D01F 6/28
    B) D01G 19/14 → D01G 19/16
    C) D01B 5/02 → D01B 5/04
    D) D01D 1/06 → D01D 1/09
**Answer:** A
**Reasoning:**
- D01B 1/14 → D01B 1/18
- D01F 2/24 → D01F 2/28 → D01F 2/30 (not D01F 2/06)
- D01G 15/76 → D01G 15/78 (not D01G 15/74)
- D01D 5/04 is parallel to D01D 5/08, not a child

**Common Ancestor:** This task can range from Memory-Light to Memory-Heavy, depending on the codes provided. It requires the model to navigate the hierarchical paths of two different codes to find their nearest shared ancestor, testing both traversal and comparison skills. Below is an example showing the highest retrieval complexity:

---

**Memory-Heavy Questions - Common Ancestor**

**Question:** Nearest common ancestor of A01B 3/421 and A01B 15/06 is:
**Options:**
    A) A01B 3/00
    B) A01B 15/00
    C) A01B
    D) A01
**Answer:** C
**Hierarchical Paths:**
- A01B 3/421 → A01B 3/42 → A01B 3/40 → A01B 3/36 → A01B 3/00 → A01B
- A01B 15/06 → A01B 15/04 → A01B 15/02 → A01B 15/00 → A01B

---

# B  ADDITIONAL RESULTS

## B.1  STRUCTURED PROMPTING AND MODEL PERFORMANCE

Figure 3 presents dumbbell plots that illustrate how structured prompting can narrow the performance gap between base and enhanced language models on hierarchical classification tasks. The plots compare accuracy across different model families and enhancement categories, such as instruction-tuned and reasoning-enhanced models. The key finding is that using structured prompts effectively reduces the performance advantage of specialized models, suggesting prompt engineering can be a powerful alternative to other methods like reinforcement learning.

# C  USE OF LLMS

We made use of LLMs to perform tasks including the polishing of our writing and the LaTeX formatting of the manuscript. It is important to note that no new ideas were introduced by these models. The sole exception is where the focus of our research, prompt optimization, made us use the extraction and analysis of the LLMs' own responses.

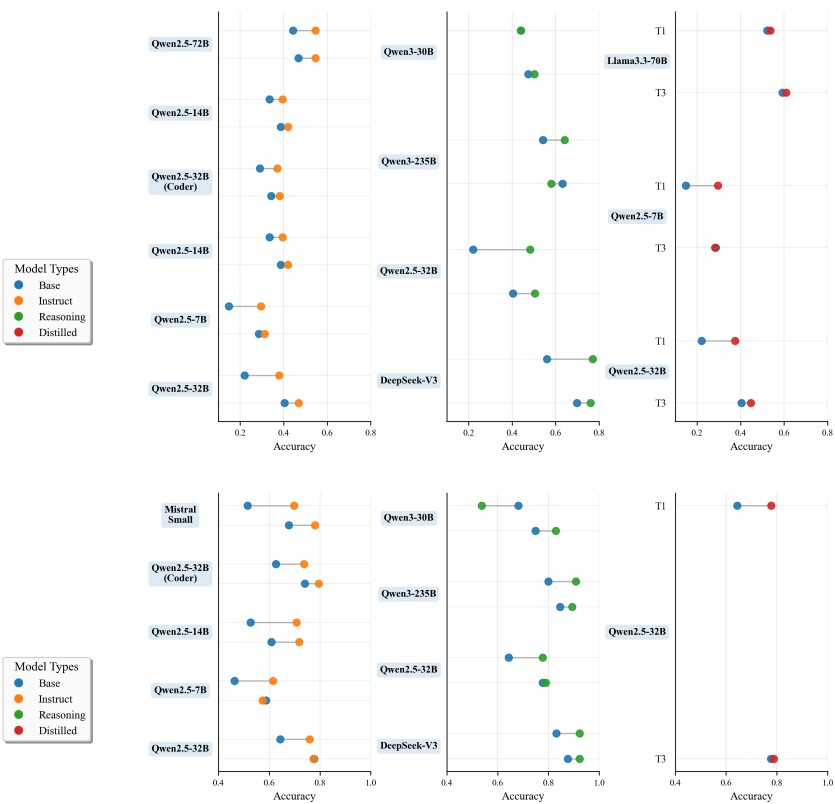

Figure 3: **Dumbbell plots revealing performance gap compression through structured prompting.** Accuracy comparison between base and enhanced models on (a) MedConceptsQA and (b) IPC codess across Templates 1 and 3. Each dumbbell connects base model performance (left endpoint) to enhanced model performance (right endpoint), with line length representing the performance gap. Model pairs span three enhancement categories: *Instruction-tuned* (Qwen2.5-7B/14B/32B/72B→Instruct, Qwen2.5-32B→Coder, Mistral-Small→Instruct), *Reasoning-enhanced* (Deepseek-V3→R1, Qwen2.5-32B→QwQ-32B, Qwen3-30B-A3B-Instruct→Thinking-2507, Qwen3-235B-A22B-Instruct→Thinking-2507), and *Distilled* (Qwen2.5-7/32B→R1-Distill-Qwen-7/32B, Llama3.3-70B-Instruct→R1-Distill-Llama-70B). The systematic compression of dumbbells from Template 1 to Template 3 demonstrates how structured prompting narrows or eliminates performance advantages of specialized models. Notably, several base models with Template 3 achieve parity or exceed their enhanced counterparts' Template 1 performance, validating prompt engineering as an alternative to RL.