# OpenReview forum: "Reinforcement Learning Improves Traversal of Hierarchical Knowledge in LLMs"
_ICLR.cc/2026/Conference — ICLR 2026 Conference Withdrawn Submission_

### Official Review · Reviewer_xBcF · 2025-10-30

**Soundness:** 1
**Presentation:** 2
**Contribution:** 1
**Rating:** 2
**Confidence:** 3

**Summary:**

The paper argues that “reasoning-enhanced” RL models outperform base models on knowledge-recall tasks (e.g., ICD/IPC code lookups) not because they store more facts, but because RL trains better hierarchical navigation at inference. The authors support this with results on MedConceptsQA and IPC: reasoning models beat base models under a simple prompt, but a structured, hierarchy-aware prompt closes most of the gap

**Strengths:**

Clear empirical observation: prompting that explicitly instructs hierarchical traversal narrows large performance gaps otherwise credited to “reasoning” training, which provides a applicable strong prompt templates

**Weaknesses:**

The central claim is quite expected and incremental relative to existing literature (e.g., “SFT Memorizes, RL Generalizes: A Comparative Study of Foundation Model Post-training”). The contributions feel too narrow for ICLR. The work consolidates a known theme rather than advancing a new method or theory.

Evidence is concentrated on two taxonomy-style benchmarks (MedConceptsQA, IPC). It is unclear whether findings hold on diverse, open-domain knowledge tasks, multi-hop QA with unstructured relations, or non-multiple-choice settings.

 The paper attributes gains in RL to *hierarchical navigation*, but structured prompts also add richer semantics, retrieval scaffolding, and option-elimination cues. The experiments don’t isolate these factors rigorously.

The paper would benefit from substantially more experimentation to support and deepen the findings. As written, the content is limited and largely observational, with insufficient analysis to justify the claims.

The writing and tables are confusing, which cannot easily be related to the work's claims.

It remains unverified what a pure memorization task is. The authors assume that retrieving information in LLMs should be very direct. But recalling needs some thinking more or less.

**Questions:**

Why are all models using the same prompting template in one column? how can that show structured prompting can substitute for RL training on hierarchical retrieval tasks?

---

### Official Review · Reviewer_rdbA · 2025-11-01

**Soundness:** 1
**Presentation:** 2
**Contribution:** 2
**Rating:** 2
**Confidence:** 4

**Summary:**

The paper challenges the conventional view that RL training improves reasoning at the cost of memorization. Through experiments on medical code (MedConceptsQA) and patent classification (IPC) tasks, the authors observe that RL-enhanced models (DeepSeek-R1, QwQ, Magistral) outperform their base counterparts by ~21 percentage points on tasks characterized as pure knowledge recall. They hypothesize this stems not from acquiring new knowledge but from learning to better navigate hierarchical structures in existing knowledge. To test this, they develop structured prompts that explicitly instruct base models to reconstruct hierarchies and perform systematic elimination, recovering ~70% of RL gains and reducing the gap to 6.1pp. The paper introduces the Hierarchical Alignment Score (HAS) metric and demonstrates that reasoning models particularly excel on tasks requiring deeper hierarchical traversal (5+ retrieval steps).

**Strengths:**

1. The reframing of RL's benefits from enhanced reasoning to improved knowledge navigation is conceptually novel
2. Within the current scope, the experimental design is comprehensive across multiple dimensions: evaluation spans several model families (DeepSeek, Qwen, Mistral, Llama) at various scales (7B-235B), includes three distinct model enhancement paradigms (instruction-tuning, reasoning-enhancement, distillation), and tests multiple prompt templates across independent runs.
3. If the core claims generalize and can be well-supported, the implication would impact positively on scientific understanding and practical deployment for RL and reasoning model development.

**Weaknesses:**

1. The paper's main claim is that RL training enhances knowledge navigation rather than logical reasoning capabilities. However, the main experiments only evaluate on tasks requiring hierarchical knowledge recall (medical codes, patent classifications). Testing exclusively on knowledge recall tasks cannot distinguish whether RL improves: (1) only knowledge navigation, (2) both knowledge navigation and reasoning, or (3) general capability that manifests in recall tasks. In order to support the claim that RL indeed does not induce new knowledge and reasoning abilities, tasks on discovering novel reasoning strategies or solutions must be used. For example, novel mathematical problem-solving, logical puzzles without factual dependencies, or algorithmic reasoning tasks requiring strategy discovery. Furthermore, the main claim requires the author to demonstrate that RL improvements on known reasoning benchmarks (AIME, SWE-Bench, etc.) can be attributed to navigation rather than reasoning, which is not tested currently.
2. Only two task types (medical/patent codes) are evaluated, both involving nearly identical hierarchical classification structures. This scope is insufficient to support broad claims about "hierarchical knowledge in LLMs" or "many benefits attributed to reasoning training" (abstract). The paper needs non-hierarchical knowledge recall tasks (e.g., factual QA, entity attributes), different types of hierarchies (taxonomies, ontologies, part-whole relationships), and tasks where structure is necessary vs. not required.
3. The claimed recall tasks, for example, medical code lookup, are framed as pure recall, but the tasks actually require semantic understanding of medical terminology and relationships. The paper doesn't distinguish between: (1) verbatim memorization, (2) semantic knowledge retrieval, and (3) structural reasoning. This conflation weakens the theoretical contribution.
4. There are a few confounding factors that should be ruled out:
- RL models may simply follow multiple-choice format instructions better.
- RL models may have better-calibrated confidence, helping with elimination.
- Reasoning models generate longer responses, so does this alone explain improvements?
- Comparison with open-ended questions suggests format matters more than navigation.

    Currently, there are no experiments systematically isolating these factors. For example, test base models with: (1) explicit chain-of-thought instructions, (2) format demonstrations, (3) few-shot examples.

**Questions:**

1. The Hierarchical Alignment Score (HAS) is computed from the ancestor sets of the predicted and gold IPC codes. If the model predicts a nearby but incorrect node, HAS can be high even when no hierarchical reasoning was performed. Moreover, HAS is computed on the code path implied by the final prediction, not on the model’s process. So, in this case, doesn't HAS mainly capture the tree proximity of outputs, not demonstrating traversal in the reasoning?

---

### Official Review · Reviewer_P9p5 · 2025-11-02

**Soundness:** 2
**Presentation:** 2
**Contribution:** 2
**Rating:** 2
**Confidence:** 4

**Summary:**

The paper shows that reasoning LLMs outperform base models on knowledge-recall. The authors hypothesize that this phenomenon is due the fact that RL encourages better hierarchical retrieval ability. The authors test this on MedConceptsQA and IPC and show that the gap is large with minimal prompting but a structured hierarchical-navigation prompt recovers most of the advantage for base models.

**Strengths:**

1. The idea about RL for LLMs improve factual recall ability by efficiently traverse hierarchical structure in the data to recall relevant information at inference time. is interesting and novel.
2. The emphasis on hierarchical structures maps well to biomedicine/genomics taxonomy. This could be relevant in practical scenarios.
3. The paper is easy to follow with detailed explanation on prompt templates, example outputs.
4. The paper raises another practical takeaway that prompting can substitute for costly RL for hierarchical lookup.
5. The claims in the paper are supported empirically (structured prompting narrows, reverse the RL advantage".)

**Weaknesses:**

1. Hierarchical recall is not unique to medicine. Many other benchmarks also require hierarchical traversal (e.g., product/category taxonomies, legal codes.) The paper should broaden coverage.
2. While the medical angle is well-motivated, the paper doesn’t clearly articulate how medical hierarchies differ from other hierarchical datasets in ways that specifically stress the hypothesized navigation skill.
3. A qualitative analysis of failure modes for the distill models is suggested since the paper lacks rationales.
4. Statistics of the results are suggested (e.g., CIs.)

**Questions:**

See the weaknesses.

---

### Official Review · Reviewer_X9Zb · 2025-11-02

**Soundness:** 3
**Presentation:** 3
**Contribution:** 2
**Rating:** 2
**Confidence:** 4

**Summary:**

This paper posits that reasoning enhanced LLMs via RL are better than base models on pure knowledge recall tasks not because they store more facts, but because RL trains hierarchical navigation at inference. A gap between Deepseek base and R1 versions is observed to this effect on MedConceptsQA and IPC codes. Distillation doesn’t transfer the navigation skill reliably (e.g., R1-Distill-Qwen-32B lags far behind R1), and math-specialized models can perform worse on hierarchy tasks.

**Strengths:**

1. Thorough literature contextualization of the various results and scoping of the problem is appreciated.
2. Creative writing and interpretation of the results. Well written and motivated from a narrative point-of-view.
3. Large breadth of models are evaluated on two medical related info retrieval settings and 1 gradations based multi-hop retrieval setting on one of the two previously mentioned datasets.

**Weaknesses:**

1. Not enough novelty and depth of experimentation and insight for the level of ICLR. The framing and intro doesn't live up to expectations in the later sections of the paper with empirical results  and could be significantly developed further. For instance, hierarchical retrieval/navigation of these LLMs, a hypothesis of how they work, isn't mechanistically or phenomenologically studied or validated.
2. The experimental scope is also constrained with respect to the setting: info retrieval in the medical domain. Other domains or even synthetic settings could be interesting to extract actual trees or graphs of the reasoning traces of these LLMs, reasoning enhanced or not.
3. There are various experimental design issues that are potentially unaddressed: RL enhancement differs across vendors, data contamination of benchmarks, prompt sensitivity. RL benefits could also come from format matching or length and not necessarily traversal.
4. The authors are probably unaware of seminal work in this area such as [Wang et. al 2024.](https://arxiv.org/abs/2402.10200) with similar propositions (CoT style reasoning is already implicit in LLMs and can be coaxed through alternative token path sampling and decoding strategies).
5. Paper feels rushed and premature. Not just with limited experimentation as raised earlier but also the citations are not properly presented/hallucinated. (e.g. lines 594-598).

**Questions:**

See above

---

### Note · Authors · 2025-11-18

I have read and agree with the venue's withdrawal policy on behalf of myself and my co-authors.